# Enantioselective Cu(I)-catalyzed borylative cyclization of enone-tethered cyclohexadienones and mechanistic insights

Sandip B. Jadhav[1,2], Soumya Ranjan Dash [2,3], Sundaram Maurya[1,2], Jagadeesh Babu Nanubolu[2,4], Kumar Vanka [2,3] & Rambabu Chegondi [1,2✉]

The catalytic asymmetric borylation of conjugated carbonyls followed by stereoselective intramolecular cascade cyclizations with in situ generated chiral enolates are extremely rare. Herein, we report the enantioselective Cu(I)-catalyzed β-borylation/Michael addition on prochiral enone-tethered 2,5-cyclohexadienones. This asymmetric desymmetrization strategy has a broad range of substrate scope to generate densely functionalized bicyclic enones bearing four contiguous stereocenters with excellent yield, enantioselectivity, and diastereoselectivity. One-pot borylation/cyclization/oxidation via the sequential addition of sodium perborate reagent affords the corresponding alcohols without affecting yield and enantioselectivity. The synthetic potential of this reaction is explored through gram-scale reactions and further chemoselective transformations on products. DFT calculations explain the requirement of the base in an equimolar ratio in the reaction, as it leads to the formation of a lithium-enolate complex to undergo C-C bond formation via a chair-like transition state, with a barrier that is 22.5 kcal/mol more favourable than that of the copper-enolate complex.

[1] Department of Organic Synthesis and Process Chemistry, CSIR-Indian Institute of Chemical Technology (CSIR-IICT), Hyderabad 500007, India. [2] Academy of Scientific and Innovative Research (AcSIR), Ghaziabad 201 002, India. [3] Physical and Materials Chemistry Division, CSIR-National Chemical Laboratory, Pune 411008, India. [4] Department of Analytical and Structural Chemistry, CSIR-Indian Institute of Chemical Technology (CSIR-IICT), Hyderabad 500007, India. ✉email: rchegondi@iict.res.in

Chiral Organoboranes are valuable synthetic intermediates in organic chemistry and the C–B linkage can be converted into C–C, C–O, and C–N bonds without racemization through stereospecific 1,2-migration[1–4]. Recently, copper has become a highly efficient and cost-effective catalyst for borylative addition with the diborane reagents on various functionalities[5–7]. Several reactions have been reported on tandem borylcupration of the carbonyls[8–12], imines[13–17], alkenes[18–45], and alkynes[46–50], followed by trapping with external electrophiles. However, copper-catalyzed borylative cyclization of alkene-tethered electrophiles has been rarely studied. Most of these intramolecular Cu-catalyzed tandem cyclization reactions investigated involve C–C bond formation via the alkylcuprate addition on carbonyls[51–57], imines[58–60], and various electrophilic carbon center bearing leaving groups (Fig. 1a)[61–67]. In addition, the copper-catalyzed borylative reaction is a more convenient approach for ligand-assisted enantiocontrol cyclization[51–67]. Still, enantioselective Cu-catalyzed conjugate addition of nucleophilic boron followed by enolate trapping via aldol-addition or nucleophilic substitution is limited to not more than three reports. Lam et al.[51], and Fernández et al.[57] reported the elegant enantioselective conjugate borylation/intramolecular 1,2-addition strategy for the construction of densely functionalized fused carbocycles. Very recently, the research group of Lautens disclosed copper-catalyzed enantioselective conjugate borylation/Mannich cyclization to access enantioenriched tetrahydroquinolines[58]. However, to the best of our knowledge, Cu-catalyzed tandem conjugate borylation/intramolecular Michael addition has not yet been studied. Herein, we report the Cu-catalyzed asymmetric borylative annulation of $C_2$-symmetric enone-tethered cyclohexadienones (bis-enones) (Fig. 1b).

Over the last decade, enantioselective desymmetrization of pro-chiral cyclohexadienones has emerged as the most powerful and convenient strategy for the rapid construction of highly functionalized bicyclic frameworks in a single operation[68–85]. We have also been developing enantioselective Rh-catalyzed cyclizations of alkyne-tethered cyclohexadienones[86–93]. However, there are only two communications on the asymmetric borylative cyclization on alkene- or alkyne-tethered cyclohexadienones to give chiral cis-hydrobenzofurans. In 2013, Lin and co-workers reported Cu-catalyzed enantioselective borylative cyclization of O-tethered cyclohexadienone-containing 1,6-enynes using bis(pinacolato)diboron ($B_2pin_2$) as a borylation reagent[77]. Here, the key regioselective borylcupration was achieved through O-directing coordination of the Cu-catalyst with propargyl ether. Subsequently, they further developed Rh(III)-catalyzed borylative annulation of alkene-tethered cyclohexadienones with excellent enantioselectivity[94]. Recently, our group[89] and Lu et al.[75], investigated the reactivity of enone-tethered cyclohexadienone using the Friedel–Crafts alkylation of indole and the Rauhut–Currier reaction, respectively.

In this work, we explore the copper-catalyzed β-borylative cascade annulation of enone-tethered cyclohexadienones to access chiral bicyclic pyran and hydroquinoline scaffolds via enantioselective desymmetrization (Fig. 1c). The prevalence of these key structural motifs in a wide range of bioactive natural products highlights the emerging importance of borylative cyclization (Fig. 1d). The reaction proceeds through conjugate borylation of

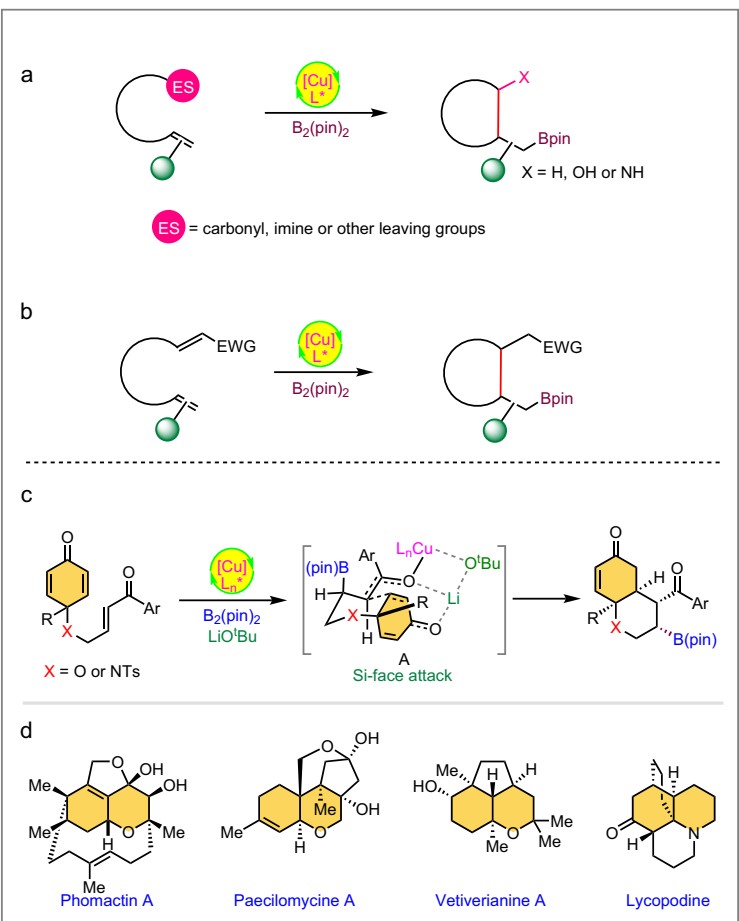

**Fig. 1 Previous and present approaches on Cu(I)-catalyzed borylative cyclization. a** Previous work: 1,2-Addition on various electrophilic sites (ES); **b** Present work: 1,4-Addtion on α,β–unsaturated carbonyl compounds; **c** Enantioselective and diastereoselective desymmetrization of cyclohexadienone-tethered enones; **d** Natural products containing bicyclic pyran and hydroquinoline backbone.

| entry[a] | ligand | base (equiv) | T °C | time | 2a + 3a yield [%][b] | er (2a)[c,d] |
|---|---|---|---|---|---|---|
| 1 | L1 | tBuONa (0.1) | 0 | 2h | 05 + 68 | - |
| 2 | L1 | tBuONa (0.5) | 0 | 2h | 12 + 59 | - |
| 3 | L1 | tBuONa (1.0) | 0 | 2h | 42 + 30 | - |
| 4 | L1 | tBuONa (2.0) | 0 | 2h | 78 + 00 | - |
| 5 | L1 | tBuOK (2.0) | 0 | 2h | 10 + 00 | - |
| 6 | L1 | Cs2CO3 (2.0) | 0 | 4h | 89 + 00 | - |
| 7 | L1 | K2CO3 (2.0) | 0 | 4h | 30 + 38 | - |
| 8 | L1 | tBuOLi (2.0) | 0 | 1h | 91 + 0 | - |
| 9 | L2 | tBuOLi (2.0) | 0 | 1h | 90 + 0 | 60:40 |
| 10 | L3 | tBuOLi (2.0) | 0 | 1h | 92 + 0 | 81:19 |
| 11 | L3 | tBuOLi (2.0) | -10 | 2h | 93 + 0 | 85:5 |
| 12 | L3 | tBuOLi (2.0) | -40 | 2h | 93 + 0 | 92:8 |
| 13 | L3 | tBuOLi (2.0) | -78 | 3h | 92 + 0 | 96:4 |
| 14 | L4 | tBuOLi (2.0) | -78 | 3h | <10 + 0 | - |
| 15 | L5 | tBuOLi (2.0) | -78 | 3h | 75 + 0 | 85:15 |
| 16 | L6 | tBuOLi (2.0) | -78 | 3h | 62 + 0 | 75:25 |
| 17 | L7 | tBuOLi (2.0) | -78 | 3h | <10 + 0 | - |
| 18 | L8 | tBuOLi (2.0) | -78 | 3h | 65 + 0 | 87:13 |

[a] Reaction conditions: **1a** (0.22 mmol), B$_2$(pin)$_2$ (64 mg, 0.25 mmol), Cu(CH$_3$CN)$_4$PF$_6$ (2 mg, 2.5 mol %), ligand (5 mol %).
[b] Isolated yields after column chromatography.
[c] Enantiomeric ratio (er) was determined by HPLC analysis using a chiral stationary phase.
[d] Observed ≥ 10:1 dr (for entries 12–18) from [1]H NMR analysis of isolated product **2a**.

**Fig. 2 Evaluation of enantioselective Cu(I)-catalyzed borylative cyclization.** Optimization of reaction conditions with various ligands and bases.

enone and subsequent intramolecular Si-face attack of chiral enolate on the cyclohexadienone ring via a six-membered chair-like transition state, as depicted in intermediate **A**, fixing the stereochemistry at the four new contiguous stereocenters. However, the major challenge of this reaction is the chemoselective 1,4-addition of nucleophilic boron on three different enone-functionalities within the starting substrate and diastereoselective C–C bond formation.

## Results and discussion
**Optimization studies**. Our studies commenced with optimization of the racemic reaction conditions using cyclohexadienone-

tethered enone **1a** as a model substrate and (rac)-BINAP **L1** (5 mol%) as ligand in THF (0.1 M) solvent at 0 °C (Fig. 2, entries 1–8). The reaction with B$_2$(pin)$_2$ (1.1 equiv) in the presence of Cu(CH$_3$CN)$_4$PF$_6$ (2.5 mol%) and NaO$^t$Bu (0.1 equiv) afforded a trace amount of the desired product **2a** along with significant quantity of uncyclized borylation product **3a** (entry 1). When the base loading was increased (2 equiv), the required bicyclic product **2a** was obtained exclusively in 78% yield (entry 4). Later, several bases such as K$^t$OBu, K$_2$CO$_3$, Cs$_2$CO$_3$, and Li$^t$OBu were subjected to borylative cyclization, and the yield of **2a** was significantly improved, especially with Li$^t$OBu (entry 8). Next, various representative chiral bisphosphine ligands (**L2**–**L8**) were

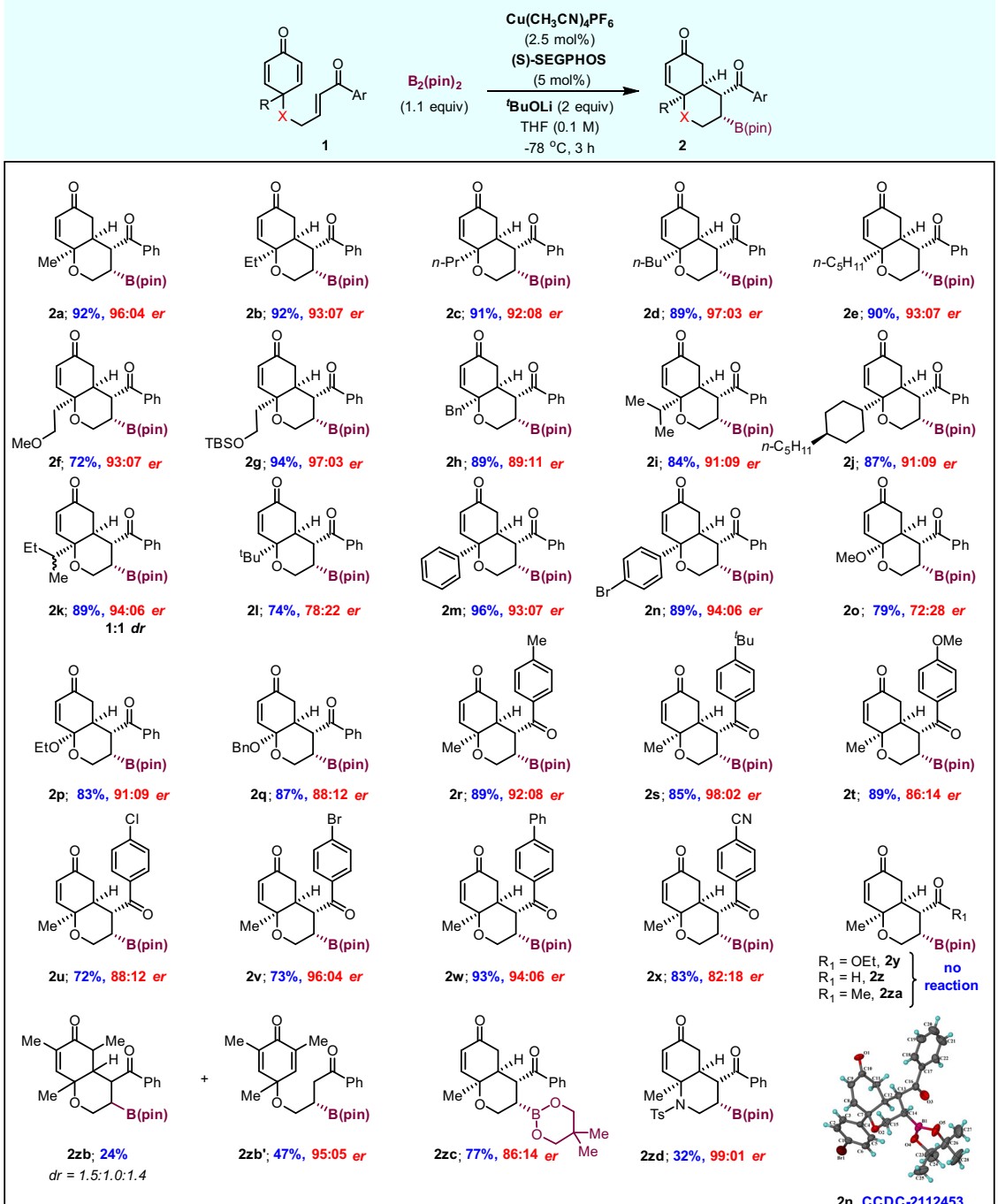

**Fig. 3 Substrate scope for *O*-tethered cyclohexadienones.** Reaction conditions: **1** (0.3 mmol), B₂(pin)₂ (84 mg, 0.33 mmol), Cu(CH₃CN)₄PF₆ (2.8 mg, 2.5 mol%), (*S*)-SEGPHOS (9 mg, 5.0 mol%), *ᵗ*BuOLi (53 μL, 0.6 mmol, 1.0 M THF solution) in THF solvent (3 mL, 0.1 M). Isolated yields after column chromatography. Enantiomeric ratio (er) was determined by HPLC analysis using a chiral stationary phase. For most of the examples, >20:1 dr was observed from ¹H NMR analysis of isolated product **2** (see ESI).

examined in the presence Li*ᵗ*OBu as a base (entries 9–18). To our delight, (*S*)-SEGPHOS controlled the enantioselectivity remarkably well, affording the bicyclic product **2a** in excellent yield (92%) with a high enantiomeric ratio at 0 °C (entry 10). Further optimization revealed that decreasing the reaction temperature (−78 °C) improved the enantiomeric ratio drastically (96:4) without significant loss of reaction yield (entry 13). However, the effect of other ligands (**L3**–**L8**) was also evaluated under the optimized conditions: either low yields or low enantioselectivity was observed in all the cases (entries 14-18). Overall, the

desymmetrization of bis-enone **1a** took place smoothly in the presence of (*S*)-SEGPHOS at −78 °C, affording the desired bicyclic product **2a** in 92% yield with 96:4 er (entry 13).

**Substrates scope.** Later, we investigated the versatility of asymmetric borylative cyclization of various enone-tethered cyclohexadienones **1** under optimized reaction conditions (Fig. 3). A variety of alkyl groups at the substrate's prochiral quaternary center gave the corresponding bicyclic enones **2a**–**2e** in 89–92% yields with >92:08 er and excellent

diastereoselectivities (>20:1). Moreover, methyl- and silyl-ether containing alkyl groups were also well tolerated (**2f** and **2g**). The reactions of benzyl, isopropyl, and cyclohexyl substituents at the enone quaternary center gave products **2h–2j** in 84–89% yield with higher enantiomeric ratios. In the case of the sec-butyl substituent, the product **2k** was obtained as a 1:1 ratio of inseparable diastereomers relative to the stereocenter on the

secondary butyl group. However, the sterically hindered tert-butyl group gave a lower enantiomeric ratio (entry **2l**). The reaction of substrates with aryl rings at the prochiral center was highly enantioselective with excellent yields (**2m** and **2n**). Interestingly, substrates with alkoxy substituents were also well tolerated in this reaction but with slightly lowered enantioselectivity (entries **2o–2q**). Later, the reactivity of substituents on the arylketone backbone was also evaluated. All para-substituted arylketones gave good to excellent yields, regardless of their electronic properties (entries **2r–2x**). However, it is interesting to note that the strong nature of electron-rich or electron-poor groups (OMe or CN) affected the enantiomeric ratio only slightly. Unfortunately, α,β-unsaturated olefins substituted with ester, aldehyde, and methyl ketone functionalities failed to give the desired products **2y**, **2z**, and **2za**, due to lack of sufficient electrophilicity of the enone and most of the starting material was decomposed. The borylation on the substrate with the methyl group at the α-position of the cyclohexadienone ring afforded the required product **2zb** in 24% yield, along with a significant quantity of the uncyclized product **2zb′** due to the steric effect. The reaction was also compatible with the bis(neopentyl glycolato)diboron reagent, giving the corresponding product **2zc** in good yield with 86:14 er. However, other diboron reagents such as tetrahydroxydiboron and bis(catecholato)diboron failed to give the desired products.

**Fig. 4 Borylative cyclization. a** Borylative cyclization of C-tethered substrate **4**; **b** parallel kinetic resolution of racemic dienone **7**.

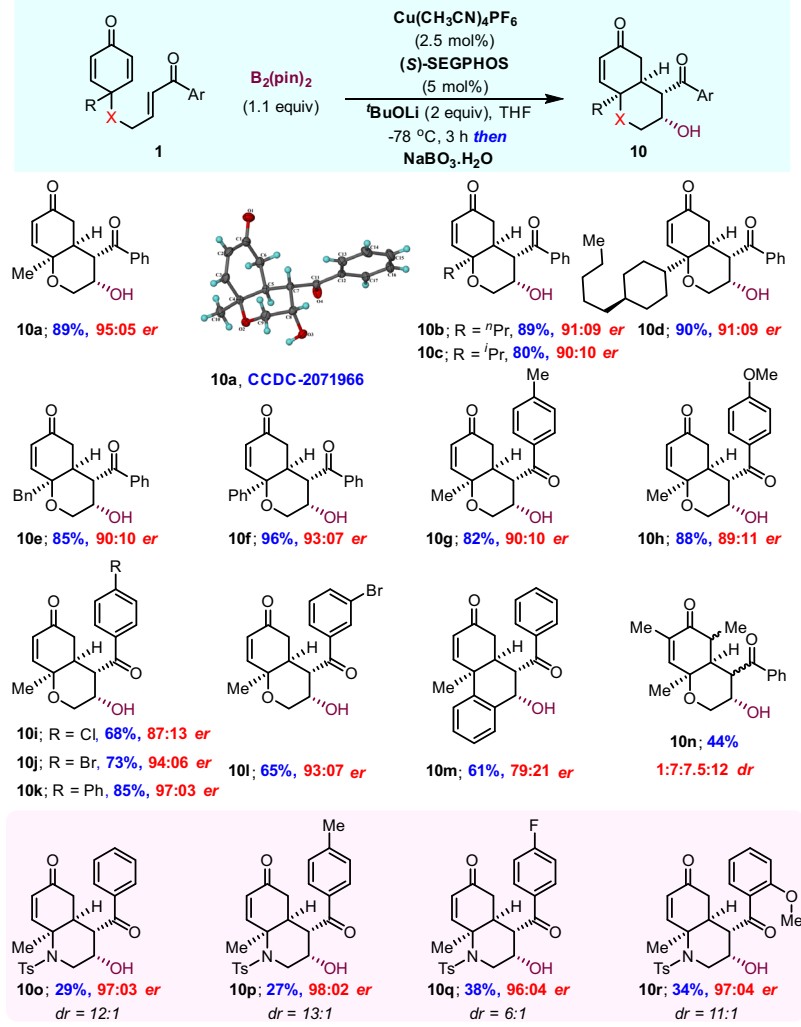

**Fig. 5 One-pot sequential borylative cyclization/oxidation.** Reaction conditions: Same as in Fig. 3 and then NaBO₃.H₂O (150 mg, 1.5 mmol, 5 equiv) was added in the same reaction flask and stirred at rt for 3 h. Isolated yields after column chromatography. Enantiomeric ratio (er) was determined by HPLC analysis using a chiral stationary phase. For all examples, >30:1 dr was observed from ¹H NMR analysis of isolated product **10**.

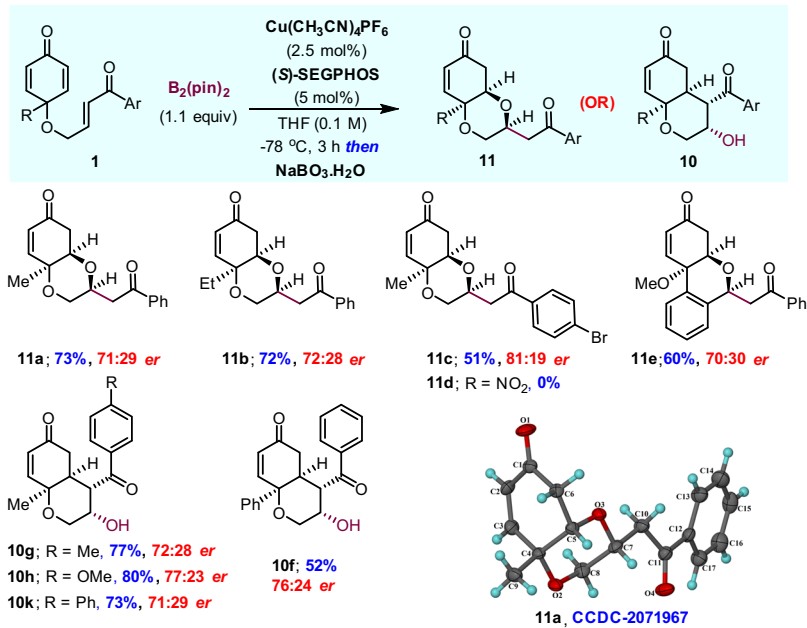

**Fig. 6 One-pot sequential borylative cyclization/oxidation without a base.** Reaction conditions: Same as in Fig. 3, without $^{t}$BuOLi base and then NaBO$_3$.H$_2$O (150 mg, 1.5 mmol, 5 equiv) was added in the same reaction flask and stirred at rt for 3 h. Isolated yields after column chromatography. Enantiomeric ratio (er) was determined by HPLC analysis using a chiral stationary phase. For all examples, exclusive diastereoselectivity was observed from $^{1}$H NMR analysis of isolated products **10** and **11**.

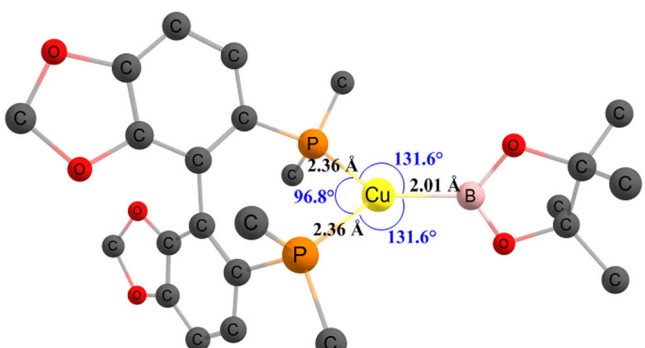

**Fig. 7 The optimized model of the boryl copper-complex [(S)-SEGPHOS-Cu-Bpin].** Hydrogen atoms have been removed for the purpose of clarity.

Interestingly, the optimized reaction conditions were also compatible with the NTs-linked substrate to provide the desired product **2zd** in excellent enantioselectivity, albeit with a low yield. We also found that the yield of the borylated product significantly drops and that it has moderate stability during purification via silica gel column chromatography. The relative syn-syn-syn-syn configuration and absolute stereochemistry of the bicyclic enone **2n** were unambiguously determined through X-ray crystallography, and the stereochemistry of all products was assumed by analogy (Fig. 3).

Next, we investigated the reactivity of the C-tethered enone **4** under standard reaction conditions (Fig. 4a). The borylative cyclization still proceeded to give the annulation product **5** as a 2:3 ratio of inseparable diastereomers in 47% yield with 91:9 er and 85:15 er, respectively. In addition, the β-borylation/aromatization product **6** was also observed in 35% yield with 80:20 er. The formation of diastereomers and aromatization probably occurred due to the absence of the Thorpe-Ingold effect during C–C bond formation. Interestingly, in the case of the racemic cyclohexadienone **7** as a substrate, the annulation product **8** was observed in

41% yield with 96:4 er from only 50% of (R)-**7** isomer via parallel kinetic resolution (Fig. 4b). However, we were unable to isolate product **9** from the (S)-isomer, and decomposition was observed.

Encouraged by the results discussed above, we sought to evaluate the one-pot sequential borylative cyclization/oxidation (Fig. 5). The Cu-catalyzed borylation of enone **1** under standard reaction conditions followed by the sequential addition of the sodium perborate oxidizing agent in the same flask afforded the β-alcohol product **10** via the β-borylation intermediate **2**. All reactions proceed with complete retention of stereochemistry, from the C–B bond to the C–O bond in a highly enantioselective fashion, and afforded the corresponding products **10a–10l** in similar yields and enantiomeric ratios, as compared to Fig. 3. The relative stereochemistry was established by single-crystal X-ray analysis of compound **10a**. It is worth mentioning that the phenyl-tethered enone was well tolerated in the annulation reaction to afford the corresponding product **10m** in 61% yield. The sequential borylation/oxidation of the substrate with methyl groups at the α-position of the cyclohexadienone ring afforded the annulation product **10n** in 44% yield in a 1:7:7.5:12 ratio of inseparable diastereomers. Regardless of the electronic properties of all substrates, we did not observe a propensity towards β-hydroxy elimination during the course of the reaction. In addition, one-pot sequential borylative cyclization/oxidation of N-tethered substrates gave the corresponding products **10o–10r** in 27–38% yield with excellent enantioselectivity and moderate to good diastereoselectivity. The low yield and high er is presumably a consequence of the steric effect from the NTs group.

Next, we investigated the sequential borylative cyclization/oxidation in the absence of base under standard conditions (Fig. 6). Interestingly, the transformation on **1a** led us to find fused dioxane **11a** via conjugate borylation/oxidation/oxa-Michael addition instead of the C-Michael adduct **10a** in 73% yield with moderate enantioselectivity[95]. It is evident that the absence of base (LiO$^{t}$Bu) has a significant effect on the enantioselectivity. The relative stereochemistry was also established by single-crystal X-ray analysis of compound **11a**. Other examples, including the enone bearing

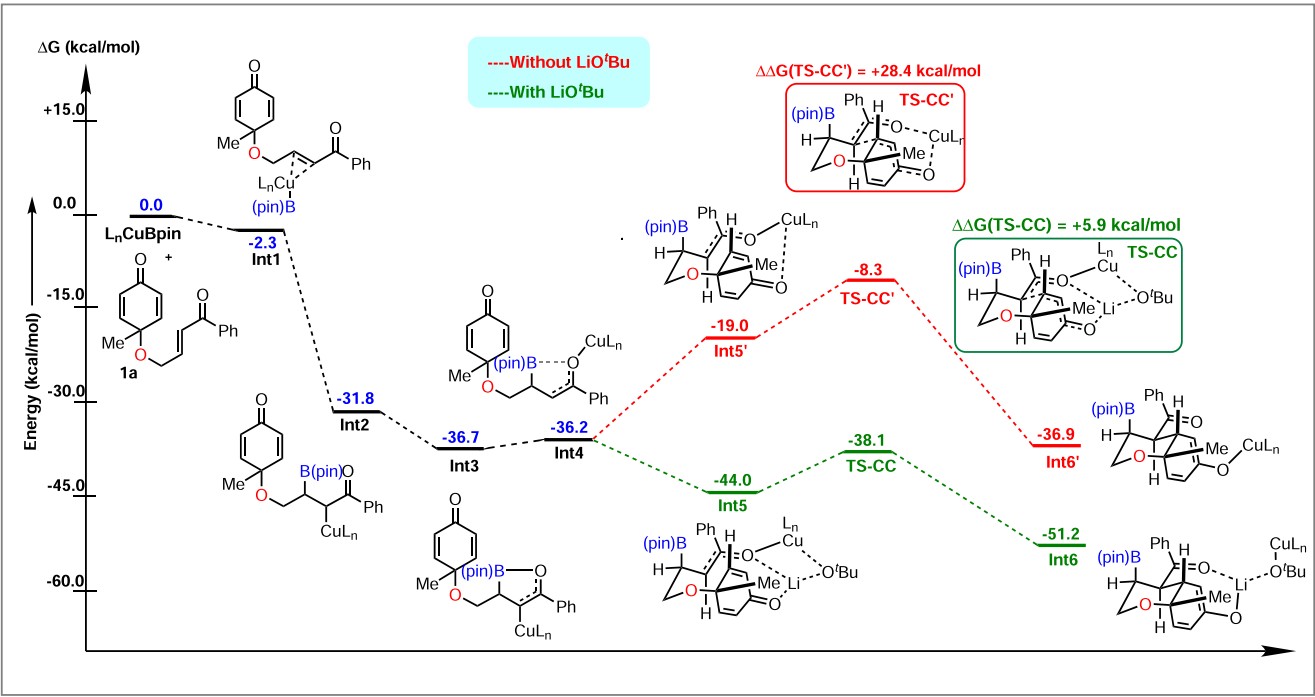

**Fig. 8 Computed free energy pathway.** All the values are in kcal/mol with respect to the separate reactants $L_nCu$-Bpin and substrate **1a**.

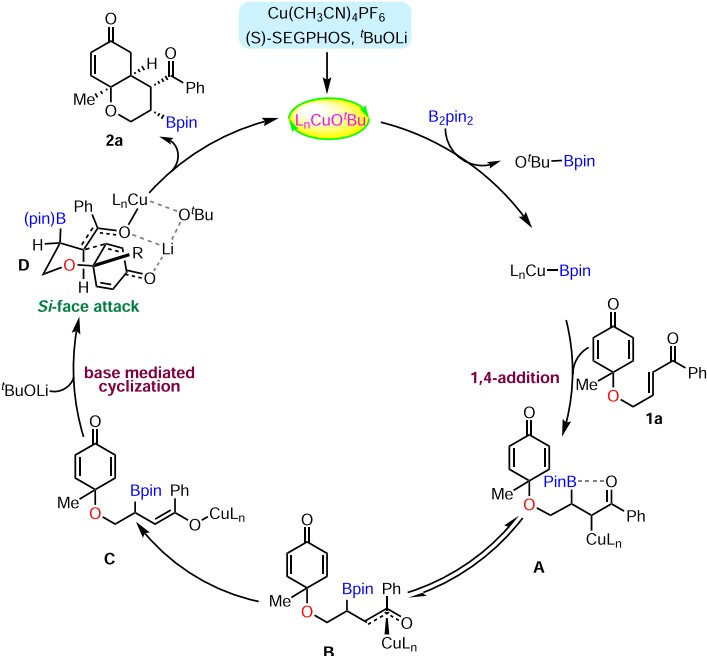

**Fig. 9 Plausible mechanism.** The more favorable lithium-enolate undergoes a Si-face attack on the cyclohexadienone ring via a six-membered chair-like transition state to fix the contiguous stereocenters.

electron-deactivating 4-bromo phenyl ring and phenyl-tethered enone, also provided oxa-Michael products with moderate er (entries **11b**, **11c**, and **11e**), along with a trace quantity (>10%) of C-Michael products. In contrast, the strong electron-withdrawing nitro group failed to give the corresponding product **11d** and the starting material was observed to decompose. Interestingly, the C-Michael addition was seen to be faster in the case of enones having the electron-rich aryl ring, affording the corresponding products **10g**, **10h**, and **10k**, albeit with a moderate enantiomeric

ratio. The substrate containing the aryl group at the quaternary carbon center also gave the C-Michael adduct **10f** in 52% yield with 76:24 er. It is very interesting to note that cyclization is more favorable than β-hydroxy elimination under these reaction conditions.

**DFT and mechanistic studies**. To understand the role of the base in the C–C bond formation and to find the factors leading to the

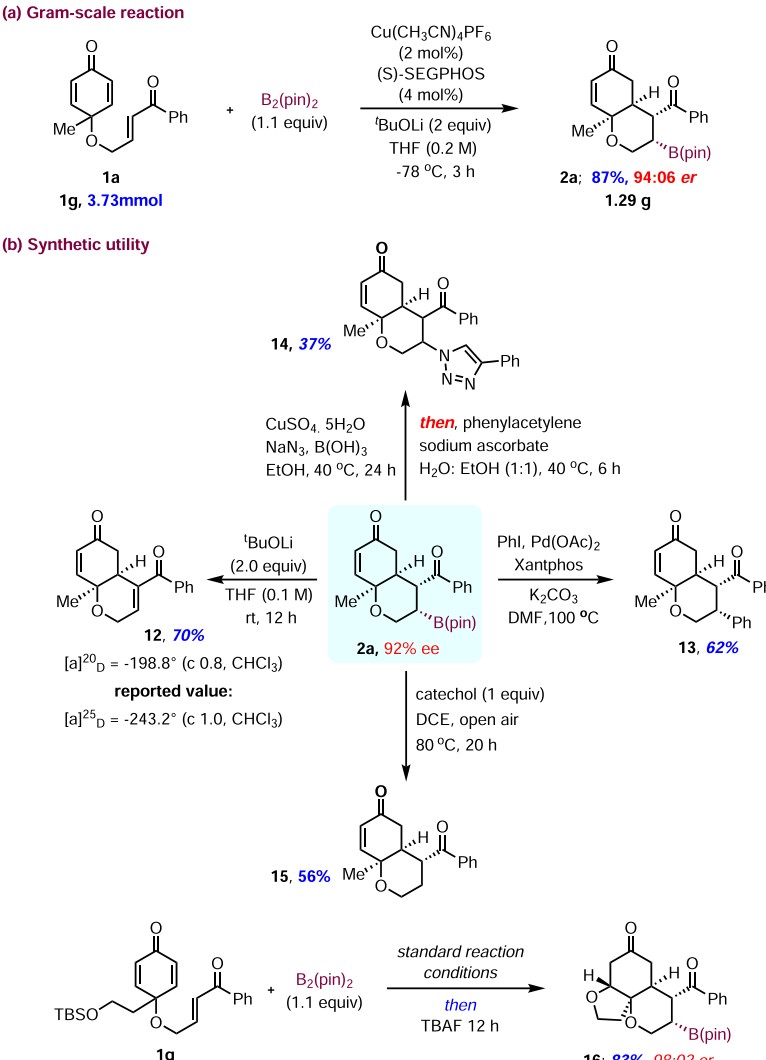

**Fig. 10 Gram-scale reaction and synthetic utility. a** Borylative cyclization of substrate **1a** in 3.73 mmol (1.0 g) scale; **b** further transformations on product **2a**; one-pot borylation/*oxa*-Michael addition of substrate **1g**.

diastereoselectivity, density functional theory (DFT) calculations were carried out using the Gaussian 09 program by employing the M06/SDD-6-311 + G(d,p)//B3LYP/SDD-6-31G(d) level of theory (details in the ESI). Before investigating the mechanistic pathway, we estimated the free energy required for the formation of the active catalyst [(S)-SEGPHOS-Cu-Bpin]. This process, which involves the addition of $B_2(Pin)_2$ to the precatalyst, leading to the formation of the boryl copper complex, was found to be exergonic ($\Delta G = -18.8$ kcal/mol). A distorted trigonal planar geometry around the coordination sphere of copper was observed for the active boryl copper complex (Fig. 7).

The mechanism determined by DFT is as follows: once the catalyst is activated, the substrate **1a** binds at the copper center to form a π-complex **Int-1**, which is 2.3 kcal/mol ($\Delta G$) more favorable compared to the reactants taken separately (Fig. 8). The π-complex then undergoes 1,4-addition along the Cu-B bond to form **Int-2**, which is highly thermodynamically favorable (**Int-2**, $\Delta G = -29.5$ kcal/mol), which immediately converts to **Int-3** ($\Delta G = -4.9$ kcal/mol), allowing the B–O interaction. This step also explains the ambiguity regarding the axial position of Bpin. Bpin forms the B–O bond by overcoming a kinetic barrier of only +5.6 kcal/mol (Supplementary Fig. 4). This stabilizes the system by 4.9 kcal/mol, as mentioned earlier. Hence, Bpin prefers the

axial position, as upon moving to the equatorial position, it will lose out on the stabilizing B–O interactions. Following the formation of **Int-3**, a rotation along the C–C(Ph)CO bond allows the carbon-bound copper to bind with oxygen, leading to the formation of **Int-4**. This was found to be almost similar in energy w.r.t to the **Int-3**, i.e, +0.5 kcal/mol ($\Delta G$). The crucial role of the extra LiO$^t$Bu was then investigated and it was found that its absence leads to a pathway where the C–C bond formation for cyclization via a copper-enolate complex (**Int-5′**) requires the overcoming of a kinetic barrier (TS-CC′) of +28.4 kcal/mol (w.r.t. to the most stable intermediate **Int-2**), which cannot be achieved at the reaction temperature (−78 °C), hence rendering the path unfavorable. However, the presence of that extra LiO$^t$Bu allows **Int-4** to form a more favorable ($\Delta\Delta G = -25.0$ kcal/mol, w.r.t **Int-5′**) lithium-enolate complex **Int-5**. This leads to a transition state where lithium is in a position to interact with all the three oxygen atoms (two from the substrate and one from the O$^t$Bu; see Supplementary Fig. 6), which plays a significant role in lowering the kinetic barrier (TS) needed for the cyclization (C–C bond formation) to +5.9 kcal/mol. This barrier is low enough that it can be overcome even at a low reaction temperature. This pathway also leads to the highly exergonic intermediate **Int-6**, where the copper complex is already in a position to be released

as the pre-catalyst ($L_nCu-O^tBu$). Figure 8 depicts the complete energy pathway as discussed and all the relevant transition states are shown in Supplementary Figs. 4 and 5 in the ESI.

Based on the above experimental results and DFT calculations, we propose the catalytic cycle shown in Fig. 9. The reaction starts with the formation of ligated copper-complex $L_nCu-O^tBu$, resulting from $Cu(CH_3CN)_4PF_6$, $^tBuOLi$, and SEGPHOS. The transmetalation of $L_nCu-O^tBu$ with $B_2pin_2$ forms the active species $L_nCu-B(pin)$, which undergoes borocupration of enone **1a** via 1,4-addition to produce the alkyl-copper(I) complex **A**. Subsequently, Intermediate **A** equilibrates with the oxa-π-allylcopper species **B** to produce the reactive copper-enolate **C**, which does not undergo C–C bond formation due to the high kinetic barrier ($\Delta\Delta G^{\#} = +28.4$ kcal/mol). As a result, transmetalation of intermediate **C** with excess LiO$^tBu$ provides the lithium-enolate **D** and regenerates the active catalyst $L_nCu-O^tBu$. The more favorable ($\Delta\Delta G = -25.0$ kcal/mol) lithium-enolate **D** further undergoes a Si-face attack on the cyclohexadienone ring via a six-membered chair-like transition state to fix the contiguous stereocenters, as depicted in product **2a**.

**Synthetic utility**. In order to demonstrate the synthetic utility, we carried out a gram-scale reaction on **1a** with a slightly reduced catalyst loading under standard conditions (Fig. 10a). A similar range of enantioselectivity of **2a** was observed with 87% yield and it was further subjected to base mediated elimination to afford the known bicyclic enone **12**[75] in 70% yield (Fig. 10b). The absolute stereochemistry was again confirmed by the optical rotation of enone **12**, which is consistent with the X-ray crystallographic analysis of compound **2n**. In addition, the Pd-catalyzed Suzuki–Miyaura cross-coupling reaction of boronate **3a** with phenyl iodide allowed the installation of the aryl ring at the β-carbon to give the corresponding product **13** in 62% yield[96–98]. A copper-catalyzed one-pot in situ azidation/ [3 + 2]-cycloaddition of boronate **2a** with $NaN_3$ and phenylacetylene provided 1,2,3-triazole **14** in moderate yield[99]. To further highlight the importance of this method, organoborane **2a** was converted to the corresponding alkane **15** via a mild radical-mediated C–B cleavage, with simple catechol in the presence of open air[100]. One-pot sequential borylative cyclization of silyl-ether **1g** under standard conditions, followed by addition of TBAF in the same reaction flask afforded the tricyclic product **16** in 83% yield and 98:2 er with exclusive diastereoselectivity via the desilylation/oxa-Michael reaction.

In summary, we have developed the enantioselective Cu(I)-catalyzed β-borylation/Michael addition of prochiral enone-tethered 2,5-cyclohexadienones. The reaction proceeds via 1,4-borocupration at the enone followed by a Si-face attack of chiral enolate on the cyclohexadienone ring via a chair-like transition state. DFT calculations explain the requirement of the excess base, leading to the formation of the more favorable chiral lithium-enolate, which undergoes C–C bond formation in the key desymmetrization step. One-pot sequential borylation/cyclization/ oxidation afforded the corresponding alcohols without affecting the yield and enantioselectivity. This asymmetric desymmetrization strategy has broad substrate scope, generates highly functionalized bicyclic enones bearing four contiguous stereocenters with excellent yield, enantioselectivity, and diastereoselectivity offers new prospects in the rapid synthesis of highly functionalized structural motifs. The synthetic utility of this reaction has been demonstrated with a gram scale reaction and further chemoselective transformations on the product **2a**. Further studies on the related asymmetric cyclizations of prochiral cyclohexadienones are underway in our laboratory.

## Methods

**General procedure for the borylative cyclization reaction**. A solution of $Cu(CH_3CN)_4PF_6$ (2.8 mg, 2.5 mol%), (S)-SEGPHOS (9 mg, 5 mol%), $B_2$(pin)$_2$ (84 mg, 0.33 mmol), and $t$-BuOLi (53 µl, 0.6 mmol, 1 M in THF) in dry THF (2.0 mL) was stirred at room temperature for 15 min and then maintained at −78 °C. A solution of enone **1** (0.3 mmol) in dry THF (1.0 mL) was added via syringe and the resulting mixture was stirred at −78 °C for 3 h. The reaction mixture was quenched with saturated $NH_4Cl$ (10 mL) solution and extracted with EtOAc (3 × 15 mL) and dried over anhydrous $Na_2SO_4$, filtered, and concentrated in vacuo. The resultant crude product **2** was purified by column chromatography (hexanes/EtOAc).

**General procedure for the one-pot sequential borylative cyclization/oxidation reaction**. A solution of $Cu(CH_3CN)_4PF_6$ (2.8 mg, 2.5 mol%), (S)-SEGPHOS (9 mg, 5 mol%), $B_2$(pin)$_2$ (84 mg, 0.33 mmol), and $^tBuOLi$ (0.53 µl, 0.6 mmol, 1 M in THF) in dry THF (2.0 mL) was stirred at room temperature for 15 min and then maintained at −78 °C. A solution of enone **1** (0.3 mmol) in dry THF (1.0 mL) was added via syringe and the resulting mixture was stirred at −78 °C for 3 h then $NaBO_3·H_2O$ (150 mg, 1.5 mmol) in $H_2O$ (2 mL) was added in one portion and the resulting mixture was stirred vigorously at room temperature for 3 h under open air. The reaction mixture was quenched with saturated $NH_4Cl$ solution (10 mL) and extracted with EtOAc (3 × 15 mL) and dried over anhydrous $Na_2SO_4$, filtered, and concentrated in vacuo. The resultant crude product **10** was purified by column chromatography (hexanes/EtOAc).

## Data availability

The authors declare that the data supporting the findings of this study are available within the article and Supplementary Information file, or from the corresponding author upon request. The X-ray crystallographic coordinates for structures reported in this study have been deposited at the Cambridge Crystallographic Data Centre (CCDC), under deposition numbers CCDC 2112453 (2n), 2071966 (10a), and 2071967 (11a). These data can be obtained free of charge from The Cambridge Crystallographic Data Centre via www.ccdc.cam.ac.uk/data_request/cif. Source data are provided with this paper.

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

## Acknowledgements

We gratefully acknowledge the SERB, DST New Delhi, India (EMR/2017/001266), and the CSIR-IICT for financial support. S.B.J. and S.M. thank UGC, New Delhi for research fellowships. IICT Communication Number for this manuscript is IICT/Pubs./2021/085. The support and the resources provided by 'PARAM Brahma Facility' under the National Supercomputing Mission, Government of India at the Indian Institute of Science Education and Research (IISER), Pune are gratefully acknowledged.

## Author contributions

S.B.J. and S.M. performed all the experiments. S.R.D. and K.V. carried out the DFT calculations. J.B.N. carried out the X-ray crystallographic analysis. R.C. supervised and directed the project, and wrote the paper with the assistance of co-authors.

## Competing interests

The authors declare no competing interests.

## Additional information



