## [Peer Review File · Nature Communications]

Enantioselective Cu(I)-Catalyzed Borylative Cyclization of Enone-Tethered Cyclohexadienones: Scope and Mechanistic InsightsREVIEWER COMMENTS

Reviewer #1 (Remarks to the Author):

Chegondi et al.'s work reports a Cu(I)-catalyzed enantioselective β borylation/Michael addition cascade reaction on prochiral enone-tethered 2,5-cyclohexadienones. It seems that the reactivity of the two types enone functional groups in the substrates relies on O-directed coordination, as is reported by Lin et al. (JACS, 2013, 11700). There are many reports, partially included and commented in the manuscript, about the capture of the enolate intermediates formed from addition of borylcopper to α,β -unsaturated compounds by intramolecular or intermolecular electrophiles. The interesting point of the manuscript is the role of bases, which is explained by experimental and calculation methods. Therefore, the present work does not present any conceptually interesting progresses on borylcopper chemistry, and the referee recommends the manuscript be published in a more specialized journal.

Reviewer #2 (Remarks to the Author):

Chegondi et al. present a copper-catalyzed enantioselective borylative cyclization cascade process featuring the use of a tethered α,β -unsaturated carbonyl system as the initial point of borylation, and a cyclohexadienone Michael system as the terminal electrophile for the cyclization. Although borylative 1,2-difunctionalization processes are well-explored, applications leveraging enantioselective borylative additions across α,β -unsaturated ketones and other Michael systems remain rare. The reported transformation is further notable due to the use of a Michael system as both the initial point of borylcupration, and for the final cyclization step. Thus, the work has both aspects of novelty, as well as complexity. The substrate scope is sufficient. I would support publication in Nature Communications after the authors strengthen the manuscript by addressing the following points:

1) In Fig 3 the authors found that the formation of the desired cyclized product is inhibited when methyl groups are present at the α -positions of the cyclohexadienone, and instead the non cyclized formal hydroboration product 2zb is formed. However, in Fig 6, when the authors add an oxidant after the completion of the reaction, product 10n is formed – implying that the reaction indeed proceeds when methyl groups are present at the α -positions of the cyclohexadienone. How come the authors observed only the hydroboration product previously, but then with the help of an oxidation were able to generate the corresponding alcohol? If the β -hydroxy products are more stable and easily isolable than the corresponding β -boryl products, they should then attempt to inspect the reaction to generate 2zb – by checking crude NMR if the desired cyclization product has been formed or otherwise by isolating any decomposition products to indicate why they could not isolate the desired borylated cyclization product. Regardless, the authors' discussion and reasoning on why the uncyclized 2zb is favored is not quite precise and should be reworded with more explanation or detail added.

2) In Fig 7, the authors show oxa-Michael products wherein the Bpin acts as a pronucleophile – and oxidation enables the subsequent nucleophilic attack of the hydroxyl moiety. Conceptually, this resembles a previous report which the authors might want to cite (Org Lett. 2021, 2720-2725).

3) The authors present DFT calculations in calculating the reaction. In one of the described pathways, the copper catalyst, initially ligated to the oxygen of the tethered Michael acceptor, migrates and becomes ligated to the cyclohexadienone ketone. Has the opposite coordination been investigated – the Cu remains coordinated to the tethered Michael moiety, with the cyclohexadienone being activated by a coordinated lithium.

4) Relatedly, the authors proposed the formation of the O-bound copper enolate before the cyclization. However, this could very possibly be detrimental to the diastereoselectivity of the cyclization. Have the authors any insight into the possibility of the carbon-bound copper enolate

acting as the nucleophile, presumably via a migratory insertion – as opined in the previous point, the activation of the cyclohexadienone system could then instead be due to lithium ion coordination at the cyclohexadienone ketone.

5) The authors demonstrated the synthetic utility of the C-B bond containing products via two different reactions: an elimination to reform the Michael acceptor, and a very traditional Suzuki reaction with Crudden conditions. In this reviewer's opinion this is insufficient, especially for a broad readership journal such as Nature Communications, and the authors should do more derivatization studies to further demonstrate the utility of the installed C-B bond.

6) The manuscript contains a large number of typographical errors, as well as inconsistencies in the figures (for e.g. Me vs CH₃) – a careful reading is necessary, and would improve the manuscript.

Reviewer #3 (Remarks to the Author):

The authors described an asymmetric copper catalyzed borylative cupration/1,4-addition cascade reaction of aryl enones with tethered cyclohexadienones to construct functionalized bicyclic enones with ee% values and yields. Relative borylative cupration of conjugated alkenes followed by intramolecular addition to imminyl or carbonyl group have been reported. And analogous transformations of alkynes tethered cyclohexadienones have also been realized. The current work is kind of integrative innovation, namely both alkene boron acceptor and the organocuprate interceptor are both enones, albeit are different types. The selective borylative cupration of the aryl enone over the cyclic cyclohexadienone is the key point for the success of this chemistry. The substrate scope is quite narrow, seems only applicable for ether tether, and aryl enones. I am quite curious if N or S linker is viable in place of the O as the linking atom. The author should also make comments on the usefulness of the product therein formed in organic synthesis or biologic sciences.

Reviewer #4 (Remarks to the Author):

In this work, Chegondi and co-workers devised and developed an enantioselective Cu(I)-catalyzed β -borylation/Michael addition of prochiral enone-tethered 2,5-cyclohexadienones and presented a plausible reaction mechanism of enantioselective borylative cyclization of a range of enone-tethered cyclohexadienones by copper(I) boryl complexes using DFT combining with experimental data. The current hybrid method is worth encouraging. Given the overall assessment of the work, I'd like to recommend their manuscript for publication in Nature Communications. However, the results and discussion about the reaction mechanism is too preliminary and ambiguous. The importance and origin of the high enantioselectivity for this asymmetric catalysis did not be presented. I found the following points should be addressed to make the manuscript better.

(1) The PESs presented in this manuscript are incomplete for many steps. How are int1 and int2 connected? Similarly, how are the following pairs of species: int2 and int3, int3 and int4, int5 and final product connected? The corresponding transition states should be supplemented to help readers have a better understanding of these transformations and their relative energies.

(2) The authors described the high enantioselectivity is attributed to the different C-H... π interaction between Bpin (axial and equatorial positions) and the phenyl group. This kind of interaction region is far away from the Cu chiral ligand, so how does the essential chiral ligand affect the enantioselectivity?

(3) The authors mentioned many times that the π -complex then undergoes 1,4-addition along the Cu-B bond, but it's more like 1,2 addition from the reaction process.

(4) The R group should be methyl in Figure 9.

(5) "B3LYP/SDD-6-31G(d)//M06/SDD-6-311+G(d,p)" should be changed to "M06/SDD-6-311+G(d,p)//B3LYP/SDD-6-31G(d)".

Reviewer #5 (Remarks to the Author):

Crystallographic data look fine. Although not perfect data, the structures are refined adequately and their assignment unambiguous.

REVIEWER COMMENTS

A POINT-BY-POINT RESPONSE TO REVIEWER COMMENTS

Manuscript ID: NCOMMS-21-21437

Title: Enantioselective Cu(I)-Catalyzed Borylative Cyclization of Enone-Tethered Cyclohexadienones: Scope and Mechanistic Insights

Author(s): Sandip B. Jadhav, Soumya Ranjan Dash, Sundaram Maurya, Jagadeesh Babu Nanubolu, Kumar Vanka and Rambabu Chegondi*

Dear Reviewers,

Thank you very much for your suggestions. We have carefully revised this manuscript based on your valuable comments. The corrections are given in the revised manuscript in detail and have been highlighted in yellow. The detailed revisions described below, in the responses to the reviewers' comments:

Reviewer #1 (Remarks to the Author):

Chegondi et al.'s work reports a Cu(I)-catalyzed enantioselective β borylation/Michael addition cascade reaction on prochiral enone-tethered 2,5-cyclohexadienones. It seems that the reactivity of the two types enone functional groups in the substrates relies on O-directed coordination, as is reported by Lin et al. (JACS, 2013, 11700). There are many reports, partially included and commented in the manuscript, about the capture of the enolate intermediates formed from addition of borylcopper to α,β -unsaturated compounds by intramolecular or intermolecular electrophiles. The interesting point of the manuscript is the role of bases, which is explained by experimental and calculation methods. Therefore, the present work does not present any conceptually interesting progresses on borylcopper chemistry, and the referee recommends the manuscript be published in a more specialized journal.

Response: First of all, we thank the reviewer for taking the time and effort to review our manuscript. The enantioselective Cu(I)-catalyzed β -borylation/Michael addition on prochiral enone-tethered 2,5-cyclohexadienones has been reported for the first time. Although borylative 1,2-difunctionalization processes are well-explored, applications leveraging enantioselective borylative additions across α,β -unsaturated ketones and other Michael systems remain rare. One-pot borylation/cyclization/oxidation via the sequential addition of sodium perborate reagent afforded the corresponding alcohols without affecting yield and enantioselectivity. We have also explained the role of an equivalent molar ratio of base, and relative stereochemistry of product using DFT calculations. In addition, we have now expanded the substrate scope with N-tethered substrates with excellent enantioselectivity and synthetic utility in the revised manuscript. Based on these points, we believe that this revised manuscript is suitable for publication in Nature communications.

Reviewer #2 (Remarks to the Author):

Chegondi et al. present a copper-catalyzed enantioselective borylative cyclization cascade process featuring the use of a tethered α,β -unsaturated carbonyl system as the initial point of borylation, and a cyclohexadienone Michael system as the terminal electrophile for the cyclization. Although borylative 1,2-difunctionalization processes are well-explored, applications leveraging enantioselective borylative additions across α,β -unsaturated ketones and other Michael systems remain rare. The reported transformation is further notable due to the use of a Michael system as both the initial point of borylcupration, and for the final cyclization step. Thus, the work has both aspects of novelty, as well as complexity. The substrate scope is sufficient. I would support publication in Nature Communications after the authors strengthen the manuscript by addressing the following points:

Response: Thank you very much for taking valuable time and effort to review our manuscript and for providing positive comments and suggestions of our work.

Question 1) In Fig 3 the authors found that the formation of the desired cyclized product is inhibited when methyl groups are present at the α -positions of the cyclohexadienone, and instead the non cyclized formal hydroboration product 2zb is formed. However, in Fig 6, when the authors add an oxidant after the completion of the reaction, product 10n is formed – implying that the reaction indeed proceeds when methyl groups are present at the α -positions of the cyclohexadienone. How come the authors observed only the hydroboration product previously, but then with the help of an oxidation were able to generate the corresponding alcohol? If the β -hydroxy products are more stable and easily isolable than the corresponding β -boryl products, they should then attempt to inspect the reaction to generate 2zb – by checking crude NMR if the desired cyclization product has been formed or otherwise by isolating any decomposition products to indicate why they could not isolate the desired borylated cyclization product. Regardless, the authors' discussion and reasoning on why the uncyclized 2zb is favored is not quite precise and should be reworded with more explanation or detail added.

Response: Thank you very much for this comment. Based on this suggestion, we have repeated the experiment and carefully isolated all products using flash column chromatography. This reaction afforded the required cyclized product **2zb** in 24% yield along with significant quantity of uncyclized product 2zb' (*numbers changed now*). We observed that the cyclized product 2zb exhibited only moderate stability during purification via silica gel column chromatography. This result has been updated in Fig 3 of the manuscript and the corresponding ^1H NMR has been included in the ESI (page S-46).

Question 2) In Fig 7, the authors show oxa-Michael products wherein the Bpin acts as a pronucleophile – and oxidation enables the subsequent nucleophilic attack of the hydroxyl

moiety. Conceptually, this resembles a previous report which the authors might want to cite (Org Lett. 2021, 2720-2725).

Response: Thank you for the suggestion. We have cited this reference in the revised manuscript (Ref No. 96).

Question 3) The authors present DFT calculations in calculating the reaction. In one of the described pathways, the copper catalyst, initially ligated to the oxygen of the tethered Michael acceptor, migrates and becomes ligated to the cyclohexadienone ketone. Has the opposite coordination been investigated – the Cu remains coordinated to the tethered Michael moiety, with the cyclohexadienone being activated by a coordinated lithium.

Response: We appreciate the valuable insights of the reviewer. We have observed that the suggested pathway is indeed more favorable than the one proposed earlier. The energy profile (Fig 9) in the manuscript has been updated accordingly. A figure (Fig S2) depicting the direct comparison between both the pathways (A and B), as shown below, is also now added to the ESI.

Fig S2: Computed free energy pathway with relevant transition states (Continuation of Figure S1). All the values are in kcal/mol with respect to the separate reactants $L_n\text{Cu-Bpin}$ and substrate **1a**.

Question 4) Relatedly, the authors proposed the formation of the O-bound copper enolate before the cyclization. However, this could very possibly be detrimental to the

diastereoselectivity of the cyclization. Have the authors any insight into the possibility of the carbon-bound copper enolate acting as the nucleophile, presumably via a migratory insertion – as opined in the previous point, the activation of the cyclohexadienone system could then instead be due to lithium ion coordination at the cyclohexadienone ketone.

Response: We understand the concern raised by the reviewer. However, based on experimental evidence and computational modeling, we can say that the migratory insertion of the carbon-bound copper enolate is not possible, because of the immense steric crowding, which was also observed during the attempt to create a model for the plausible geometry.

Question 5) The authors demonstrated the synthetic utility of the C-B bond containing products via two different reactions: an elimination to reform the Michael acceptor, and a very traditional Suzuki reaction with Crudden conditions. In this reviewer’s opinion this is insufficient, especially for a broad readership journal such as Nature Communications, and the authors should do more derivatization studies to further demonstrate the utility of the installed C-B bond.

Response: Based on the concern raised by the reviewer, we have expanded the synthetic utility of the installed C-B bond and the result have been included in manuscript Fig.11. The manuscript has been revised accordingly.

We have also attempted several other transformations on product **2a**. However, this did not lead to fruitful results. These details are included below for your reference.

Question 6) The manuscript contains a large number of typographical errors, as well as inconsistencies in the figures (for e.g. Me vs CH₃) – a careful reading is necessary, and would improve the manuscript.

Response: We have revised the manuscript accordingly with the help of a native English speaker.

Reviewer #3 (Remarks to the Author):

The authors described an asymmetric copper catalyzed borylative cupration/1,4-addition cascade reaction of aryl enones with tethered cyclohexadienones to construct functionalized bicyclic enones with ee% values and yields. Relative borylative cupration of conjugated alkenes followed by intramolecular addition to imminyl or carbonyl group have been reported. And analogous transformations of alkynes tethered cyclohexadienones have also been realized. The current work is kind of integrative innovation, namely both alkene boron acceptor and the organocuprate interceptor are both enones, albeit are different types. The selective borylative cupration of the aryl enone over the cyclic cyclohexadienone is the key point for the success of this chemistry. The substrate scope is quite narrow, seems only applicable for ether tether, and aryl enones. I am quite curious if N or S linker is viable in place of the O as the linking atom. The author should also make comments on the usefulness of the product therein formed in organic synthesis or biologic sciences.

Response: Thank you very much for taking valuable time and effort to review our manuscript and give positive comments and suggestions on our work. Interestingly, the optimized reaction conditions are also compatible with the NTs-linked substrate to provide the desired product in excellent enantioselectivity, albeit with a low yield. We also found that the yield of borylated

product is substantially reduced and that the stability is moderate during purification via silica gel column chromatography. This result has been included in Fig. 3 (entry 2zd). In addition, one-pot sequential borylative cyclization/oxidation of *N*-tethered substrates gave the corresponding products **10o-10r** in 27-38% with excellent enantioselectivity and moderate to good diastereoselectivity. The low yield and high *er* are, presumably a consequence of the steric effect of the NTs group. These results have been included in the manuscript (Fig. 6).

The key chiral bicyclic pyran and hydroquinoline scaffolds, which are generated from this method, are found in a wide range of bioactive natural products highlights the emerging importance of borylative cyclization. A few related natural products have been included in Fig.1.

Reviewer #4 (Remarks to the Author):

In this work, Chegondi and co-workers devised and developed an enantioselective Cu(I)-catalyzed β -borylation/Michael addition of prochiral enone-tethered 2,5-cyclohexadienones and presented a plausible reaction mechanism of enantioselective borylative cyclization of a range of enone-tethered cyclohexadienones by copper(I) boryl complexes using DFT combining with experimental data. The current hybrid method is worth encouraging. Given the overall assessment of the work, I'd like to recommend their manuscript for publication in Nature Communications. However, the results and discussion about the reaction mechanism is too preliminary and ambiguous. The importance and origin of the high enantioselectivity for this asymmetric catalysis did not be presented. I found the following points should be addressed to make the manuscript better.

We thank the reviewer for his/her positive comments about our work. In the revised version of the manuscript, we have endeavored to bring greater clarity in the presentation and discussion of the results, so as to show how the combined computational and experimental approach explains the high enantioselectivity obtained for the presented asymmetric catalysis.

Question (1) The PESs presented in this manuscript are incomplete for many steps. How are int1 and int2 connected? Similarly, how are the following pairs of species: int2 and int3, int3 and int4, int5 and final product connected? The corresponding transition states should be supplemented to help readers have a better understanding of these transformations and their relative energies.

Response: We agree with the concerns raised by the reviewer. We have improved the energy pathway, and the corresponding transition states added in ESI (Fig S1 and Fig S2). However, we would like to mention that transformation of Int5 (now Int6) to the final product occurs during the workup protocol, which includes quenching the reaction, followed by washing by water before purification. Hence this step has not been included in the DFT calculations.

Fig S1: Computed free energy pathway. All the values are in kcal/mol with respect to the separate reactants L_nCu -Bpin and substrate **1a**.

Fig S2: Computed free energy pathway with relevant transition states (Continuation of Figure S1). All the values are in kcal/mol with respect to the separate reactants L_nCu -Bpin and substrate **1a**.

Question (2) The authors described the high enantioselectivity is attributed to the different C-H $\cdots\pi$ interaction between Bpin (axial and equatorial positions) and the phenyl group. This kind of interaction region is far away from the Cu chiral ligand, so how does the essential chiral ligand affect the enantioselectivity?

Response: Based on new investigations in response to the previous comment, we have found that the observed axial position of Bpin is primarily due to B-O interaction (Int2-Int3-Int4), which is favorable. The boron will miss out on the B-O stabilizing interaction if Bpin moves to the equatorial position. Hence, this part along with the NCI plot has been removed from the revised manuscript. We thank the reviewer for the previous comment, which enabled us to determine this modified behavior of the Bpin. One thing that we must mention here is that the chiral phosphine ligand probably has no role in determining the diastereoselectivity. The bulky chiral ligand in the active catalyst LnCu-OtBu plays a crucial role in forming the enantioselective C-B bond, while the orientation of the B(pin) (axial or equatorial position) group dictates the other three stereocenters in the TS-CC transition state.

Question (3) The authors mentioned many times that the π -complex then undergoes 1,4-addition along the Cu-B bond, but it's more like 1,2 addition from the reaction process.

Response: Here, we have mentioned 1,4-addition of B(pin) with respect to the enone group.

Question (4) The R group should be methyl in Figure 9.

Response: We thank the reviewer for mentioning the error. Figure 9 has been updated accordingly.

Question (5) “B3LYP/SDD-6-31G(d)//M06/SDD-6-311+G(d,p)” should be changed to “M06/SDD-6-311+G(d,p)//B3LYP/SDD-6-31G(d)”.

Response: We thank the reviewer for pointing this out. The change has now been made in the manuscript.

Reviewer #5 (Remarks to the Author):

Crystallographic data look fine. Although not perfect data, the structures are refined adequately and their assignment unambiguous.

Response: Thank you. Crystals of KA1125 compound were of moderate quality. Several crystallization attempts were made in the revision time to obtain better quality crystals by changing the solvents and solvent combinations. We obtained good quality crystals in acetonitrile and hexane (1:1) solvent systems by the slow evaporation of the solvent technique. Single crystal X-ray diffraction experiment was performed and the structure was solved. The absolute configuration of the compound remains unchanged. The crystal data (CIF) was deposited at the Cambridge Structural Database and was assigned with a CCDC number 2112453. The new crystal data (KB151, CCDC 2112453) replaces the previous crystal data (KA1125, CCDC 2071968) in the revised manuscript.

REVIEWERS' COMMENTS

Reviewer #2 (Remarks to the Author):

The revised manuscript and the response to the questions has significantly improved the manuscript. I support publication.

Reviewer #3

< In private comments to the Editor, the reviewer says that they support publication. >

Reviewer #4 (Remarks to the Author):

Considering the indispensable aryl substituent of enone in all successful reactions, the steric hindrance between such aryl and the phenyl of the chiral ligand could determine enantioselectivity. Basically, the revised manuscript by Chegondi et al. satisfactorily addresses this reviewer's previous concerns. I recommend publication of the manuscript in Nature Communications.

REVIEWER COMMENTS

A POINT-BY-POINT RESPONSE TO REVIEWER COMMENTS

Manuscript ID: NCOMMS-21-21437A

Title: Enantioselective Cu(I)-Catalyzed Borylative Cyclization of Enone-Tethered Cyclohexadienones: Scope and Mechanistic Insights

Author(s): Sandip B. Jadhav, Soumya Ranjan Dash, Sundaram Maurya, Jagadeesh Babu Nanubolu, Kumar Vanka and Rambabu Chegondi*

Dear Reviewers,

Thank you very much for your suggestions.

Reviewer #2 (Remarks to the Author):

The revised manuscript and the response to the questions has significantly improved the manuscript. I support publication..

Response: We thank the reviewer for taking the time and effort to review our manuscript and for providing positive comments and suggestions of our work.

Reviewer #3

< In private comments to the Editor, the reviewer says that they support publication. >

Response: Thank you very much for taking valuable time and effort to review our manuscript.

Reviewer #4 (Remarks to the Author):

Considering the indispensable aryl substituent of enone in all successful reactions, the steric hindrance between such aryl and the phenyl of the chiral ligand could determine enantioselectivity. Basically, the revised manuscript by Chegondi et al. satisfactorily addresses this reviewer's previous concerns. I recommend publication of the manuscript in Nature Communications.

Response: We thank the reviewer for taking the time and effort to review our manuscript and for providing positive comments and suggestions of our work